# Transcriptome and Metabolome Reveal the Molecular Mechanism of Barley Genotypes Underlying the Response to Low Nitrogen and Resupply

**DOI:** 10.3390/ijms24054706

**Published:** 2023-02-28

**Authors:** Gang Wang, Juncheng Wang, Lirong Yao, Baochun Li, Xiaole Ma, Erjing Si, Ke Yang, Chengdao Li, Xunwu Shang, Yaxiong Meng, Huajun Wang

**Affiliations:** 1State Key Laboratory of Aridland Crop Science, Gansu Key Laboratory of Crop Improvement and Germplasm Enhancement, Lanzhou 730070, China; 2Department of Crop Genetics and Breeding, College of Agronomy, Gansu Agricultural University, Lanzhou 730070, China; 3Department of Botany, College of Life Sciences and Technology, Gansu Agricultural University, Lanzhou 730070, China; 4Western Barley Genetics Alliance, College of Science, Health, Engineering and Education, Murdoch University, Murdoch, WA 6150, Australia

**Keywords:** barley, genotypes, nitrogen, transcriptome, metabolome

## Abstract

Nitrogen is one of the most important mineral elements for plant growth and development. Excessive nitrogen application not only pollutes the environment, but also reduces the quality of crops. However, are few studies on the mechanism of barley tolerance to low nitrogen at both the transcriptome and metabolomics levels. In this study, the nitrogen-efficient genotype (W26) and the nitrogen-sensitive genotype (W20) of barley were treated with low nitrogen (LN) for 3 days and 18 days, then treated with resupplied nitrogen (RN) from 18 to 21 days. Later, the biomass and the nitrogen content were measured, and RNA-seq and metabolites were analyzed. The nitrogen use efficiency (NUE) of W26 and W20 treated with LN for 21 days was estimated by nitrogen content and dry weight, and the values were 87.54% and 61.74%, respectively. It turned out to have a significant difference in the two genotypes under the LN condition. According to the transcriptome analysis, 7926 differentially expressed genes (DEGs) and 7537 DEGs were identified in the leaves of W26 and W20, respectively, and 6579 DEGs and 7128 DEGs were found in the roots of W26 and W20, respectively. After analysis of the metabolites, 458 differentially expressed metabolites (DAMs) and 425 DAMs were found in the leaves of W26 and W20, respectively, and 486 DAMs and 368 DAMs were found in the roots of W26 and W20, respectively. According to the KEGG joint analysis of DEGs and DAMs, it was discovered that glutathione (GSH) metabolism was the pathway of significant enrichment in the leaves of both W26 and W20. In this study, the metabolic pathways of nitrogen metabolism and GSH metabolism of barley under nitrogen were constructed based on the related DAMs and DEGs. In leaves, GSH, amino acids, and amides were the main identified DAMs, while in roots, GSH, amino acids, and phenylpropanes were mainly found DAMs. Finally, some nitrogen-efficient candidate genes and metabolites were selected based on the results of this study. The responses of W26 and W20 to low nitrogen stress were significantly different at the transcriptional and metabolic levels. The candidate genes that have been screened will be verified in future. These data not only provide new insights into how barley responds to LN, but also provide new directions for studying the molecular mechanisms of barley under abiotic stress.

## 1. Introduction

Nitrogen is the most important mineral element in plants, the essential nucleotide and protein for life, and the main component of plant hormones [1,2]. To increase crop yields, more than 100 million tons of nitrogen fertilizer is used annually worldwide, but excessive nitrogen application may cause air and water pollution [3]. In some parts of the world, excessive nitrogen also has a negative impact on biodiversity, human health, and climate [4,5]. Simultaneously, excessive nitrogen application also promotes the ratio of environmental nitrogen to phosphorus, thus affecting ecological structure and function [6]. Moreover, plants growing under excessive nitrogen application are more likely to lodge due to overgrowth and tenderness of branches, diseases, and insect pests, thereby reducing crop quality [7]. Generally, the ratio of yield to total nitrogen supply is called nitrogen use efficiency (NUE) [8]. On average, the absorption of nitrogen fertilizer is less than half of the amount of nitrogen fertilizer applied, so it is meaningful to improve the NUE of crops [9].

Nitrate and ammonium are the two main forms of inorganic nitrogen in soil. Among them, nitrate is the main form of nitrogen available for most higher plants in the aerobic environment, and ammonium is usually the main form of plants in waterlogged or acidic soil [10,11]. The process of nitrogen utilization can be divided into absorption, transport, and assimilation, so the transporters and assimilation enzymes of nitrate or ammonium are the most important components that determine the NUE [12]. For most plants, the nitrate absorbed by the root is assimilated into the root, while the majority is transported to the ground. In general, nitrate is first reduced to nitrite by nitrate reductase (NR) in the cytoplasm of leaves, and then further reduced to ammonium by nitrite reductase (NIR) in chloroplasts and glutamine synthetase (GS) in the cytoplasm [13,14]. Ammonium, coming from nitrate or directly absorbed by the ammonium transporter, is assimilated to amino acids by glutamate synthase (GOGAT), while α-ketoglutarate acid is the product of tricarboxylic acid cycle (TCA), with a C5 carbon skeleton. α-Ketoglutaric acid and ammonia can be converted into glutamic acid under GS and GOGAT [15]. α-Ketoglutaric acid can also be converted into glutamic acid under glutamate dehydrogenase (GDH), which is the key enzyme connecting biological carbon and nitrogen metabolism [16]. Glutamate synthase includes ferredoxin-dependent glutamate synthase (FD-GOGAT) and NADH-dependent glutamate synthase (NADH-GOGAT); the former mainly exists in the chloroplast, and the latter in the cytoplasm [17].

Both the yield of barley and the planting area of cereal crops rank fourth in the world [18]. By the end of the 21st century, the annual output of barley is expected to reach 140 million tons in the world, covering an area of over 55 million hectares [19]. Barley is mainly used as animal feed and grain, as well as for malt. Currently, people are becoming more and more aware of the high nutritional value of barley, so barley is deeply loved by the public for its high content of β-glucan and low gluten [20,21]. As a highly adaptable crop, most of the barley in the world is produced in areas with poor grain growth (such as corn and rice), and barley is distributed near the arctic and subtropics. Therefore, it is of great importance to increase the yield of barley in harsh environments for the future [22]. Moreover, the great genetic diversity and resilience of barley in harsh environments, and the unique adaptation mechanism to abiotic stress, are of great value for the agroecological transformation and the reduction of nitrogen fertilizer input [23,24,25]. However, at present, there are insufficient studies on the molecular mechanism of barley tolerance to low nitrogen stress [24,26,27].

Transcriptome refers to the sum of all RNAs transcribed by a particular tissue or cell at a certain time or in a certain state. Transcriptome has been used in many plants to investigate the complex regulatory mechanisms of roots and leaves under nitrogen stress. In the transcriptome analysis of potatoes under low nitrogen stress, the co-expressed genes and potential pathways related to nitrogen transport and absorption in roots, stems, and leaves were confirmed [28]. The transcriptome of *Elymus breviaristatus* treated with different concentrations of ammonium showed that ribosomal proteins were regulated in roots and might affect the regulation of sieve tube transport or stress resistance [29]. In addition, studies have also explored the physiological and comparative transcriptome mechanism of high NUE acquisition by using a low nitrogen-tolerant genotype and a low nitrogen-sensitive genotype. For example, through transcriptome analysis of two Tibetan wild barley genotypes with different NUEs under low nitrogen, it was found that the high expression of the nitrate transporter and the response for auxin (IAA) and ethylene (ETH) to low nitrogen stress may also be related to genotypes [24]. The transcriptome of pepper genotypes with different NUEs under low nitrogen was found to be different DEGs that do not directly participate in nitrogen metabolism [30].

Metabolomics is applied to crop abiotic stress, aiming at investigating the changes of its metabolites or the changes with time after abiotic stress, thereby screening the differential metabolites (DAMs) between the experimental group and the control group, exploring the DAMs and metabolic pathways of crops after abiotic stress, and revealing the mechanism of metabolism involved in crop stress resistance. At present, multi-group analysis is widely used to study the response of plants to abiotic stress [31,32,33]. Schlüter used a combination of transcriptomes in studying the changes in carbon, nitrogen, and phosphorus metabolism in maize under low temperature and low nitrogen [34]. There are also some studies on the response of Arabidopsis roots to nitrogen and hormones, by combining transcriptome and phenotypic analyses [35,36]. Some studies also evaluate how parsley integrates nutrition and hormone signaling pathways, thereby controlling root growth and development [37]. In addition, the mechanism of the low nitrogen tolerance of wild soybean seedlings has been revealed by the analysis of soybean transcriptome and metabolome in some studies [38].

W26 and W20 are two genotypes with significant differences in NUE after low nitrogen stress, and they were screened in the field previously. After the two genotypes were treated with low stress (LN) and resupply nitrogen (RN), the dry weight and nitrogen content were measured, and then the NUEs after 21 days of plant growing for the two genotypes were estimated after LN, and it was proved that there was indeed a large difference in NUE between the two genotypes after LN. After the transcriptome and metabolomic analysis of leaves and roots, the differentially expressed genes (DEGs) and differentially expressed metabolites (DAMs) could be identified. Based on the enrichment analysis of DEGs and DAMs, the difference in metabolic pathways between the two genotypes after LN and RN was also identified. In addition, this study also focused on the differential expression of key enzyme genes and nitrogen transporter genes in the nitrogen metabolism pathway to better understand the situation of the nitrogen metabolism pathway in barley after LN stress. At present, single-omics studies on barley after low nitrogen stress are usually conducted [39,40], but there are few reports on the differences in NUEs in different parts of genotypes. This study not only provides unique access to the nitrogen reprogramming of barley under deficiency/resupply, but also demonstrates the close cooperation between nitrogen-efficient genes and metabolic functions.

## 2. Results

### 2.1. Dry Weight and Biomass under Different Nitrogen Treatments

From the appearance of the two genotypes, it can be seen that there are great differences in the morphology of W26 and W20 under different treatments (Figure 1), but only the results of shoot dry weight and root dry weight can quantify this difference. After 3 days of LN, there was a significant difference in shoot CK and LN between W26 and W20 (Figure 2a). The roots of W26 increased by 17.27% compared with the CK after LN in roots, but there was no significant difference between the CK and the LN of W20 (Figure 2b). As for biomass (the sum of shoot dry weight and root dry weight), there was no significant difference between the CK and the LN of W26 (Figure 2c), while the LN of W20 decreased by 7.52% compared with the CK. After 18 days of LN, there was still a great difference in the shoot dry weight of the two genotypes between the CK and the LN, and there was also no significant difference in root dry weight between the two genotypes after LN. W26 and W20 increased by 13.52% and 10.83%, respectively, compared with the CK. Simultaneously, there was no significant difference in biomass between the CK and the LN in W26, while W26 decreased by 17.77% compared with the CK after LN. After 21 days, there was a significant difference in the dry weight of shoots of W26 between the CK and the LN, as well as the CK and the RN. Meanwhile, there was no significant difference between the LN and the RN, but great differences among CK, LN, and RN as for W20. For root dry weight, there was no significant difference between W26 LN and RN, and the root dry weight of W20 increased by 17.62% and 15.37%, respectively compared with the CK, while the LN and RN of W20 increased by 13.07% and 9.72%, respectively. There was no significant difference in the biomass of W26 among the CK, LN, and RN, but W20 showed a significant difference among the three different treatments.

### 2.2. Nitrogen Content under Different Nitrogen Treatments

After 3 days of nitrogen stress, there was no significant difference in shoot nitrogen content between the CK and the LN of W26, but the shoot nitrogen content of W20 significantly decreased, from 39.04 mg·plant^−1^·g^−1^ to 35.95 mg·plant^−1^·g^−1^ (Figure 3a). At this time, there was no significant difference in root CK and LN between the two genotypes (Figure 3b). After 18 days of low nitrogen stress, the nitrogen content in shoots of W26 and W20 decreased by 3.69 mg·plant^−1^·g^−1^ and 6.66 mg·plant^−1^·g^−1^, respectively, and there were significant differences between CK and LN, and the nitrogen content in the roots of W26 decreased from 30.77 mg·plant^−1^·g^−1^ to 28.30 mg·plant^−1^·g^−1^, while that of W20 decreased from 30.47 mg·plant^−1^·g^−1^ to 21.07 mg·plant^−1^·g^−1^. After 21 days, there was a significant difference in nitrogen content between CK and LN of W26 shoots, but there was no significant difference between LN and RN, and there were significant differences among three different treatments of W20. The nitrogen content was 41.73 mg·plant^−1^·g^−1^, 37.83 mg·plant^−1^·g^−1^ and 37.50 mg·plant^−1^·g^−1^, respectively. Simultaneously, there was a significant difference between the CK and the LN, and it was also found between the CK and RN in the root nitrogen content of W26, but there was no significant difference between the LN and the RN. There were significant differences among the three different treatments of W20, the values were 33.97 mg·plant^−1^·g^−1^, 26.07 mg·plant^−1^·g^−1^, and 32.99 mg·plant^−1^·g^−1^.

### 2.3. NUE Estimation after 21 Days of Plant Growing

The measured dry weight and nitrogen content in the leaf and root genotypes of barley in 21 days under CK and LN were estimated. The definition of NUE itself is also very complex, and there is no fixed calculation method. N uptake efficiency (NUpE = Output nitrogen/Input nitrogen × 100%) is one of the methods to estimate NUE [17]. The NUpE can be estimated according to the nitrogen content, dry weight, and seeding growth conditions (nutrient solution concentration, planting density, nutrient solution replacement times). After 21 days of treatment, the NUE of the two genotypes increased significantly under the LN condition (Table 1). The NUE of W26 in the LN condition is 87.52%, which was significantly higher than 61.74% of W26.

### 2.4. Transcriptome Profiling under Different Nitrogen Treatments

#### 2.4.1. Results of RNA-Seq

The transcriptome data of the 72 samples described in this study have been stored in the National Biotechnology Information Center (NCBI) database, with the biological project entry number PRJNA896249. The total number of bases of 497.49 Gb raw data was obtained by sequencing, with a total of 3,316,251,936 read numbers. After filtering, the Q20 values of GC were all greater than 96.26%, and those of Q30 were all greater than 90.52%. The percentages of G and C in the four bases after filtering were 47.3% and 58.76%, respectively. The summary of the data with an overall sequencing error rate of less than 0.03% is listed in Appendix A, which met the sequencing quality control requirements. The Pearson correlation coefficient among the three biological repeats was higher (Appendix A), which can be analyzed and sequenced later. The minimum number of clean reads in all samples was 40,028,848, with the maximum 52,892,072. The number and percentage of reads aligned to the genome were 72.26–92.78%. Among these genes that can match the genome, the probability of specific pairing was 71.12–89.5% (Appendix A).

#### 2.4.2. Selection of DEGs under Different Nitrogen Treatments

In the leaves, the numbers of up-regulated and down-regulated genes in W26 were all less than those in W20 on the 3rd days and 21st days (Figure 4a–d). On the 18th day, the numbers of up-regulated and down-regulated genes in W26 (1782 and 1624, respectively) were much greater than those in W20 (374 and 1071, respectively), which may be related to the time of LN stress. However, once the normal nitrogen was restored, the number of DEGs in both genotypes rapidly declined, but it was more rapidly declined in W26. The number of W26 DEGs decreased from 3406 (1782 up-regulated and 1624 down-regulated) on the 18th day to 46 (26 up-regulated and 20 down-regulated) on the 21st day, and the number of W20 DEGs decreased from 1445 (374 up-regulated and 1071 down-regulated) on the 18th day to 301 (118 up-regulated and 183 down-regulated) on the 21st day. The reason for the above situation may be that the number of DEGs for W26 treated with LN was higher than that of the resupplied nitrogen, the low nitrogen stress effect disappears, and the number of DEGs decreases rapidly.

According to the identification of DEGs in roots, the numbers of up-regulated and down-regulated genes in W26 were greater than those in W20 on the 3rd and 21st days (Figure 4f–h). On the 18th day, the numbers of up-regulated and down-regulated differential genes in W26 were less than those in W20. The roots were the main organ for higher plants to absorb nitrogen, and were more sensitive to the change in nitrogen concentration. Moreover, nitrogen absorbed by barley roots must be transported to the shoots through the xylem and phloem. Moreover, the plant itself was much smaller in the early stage of stress, with a relatively small demand for nitrogen. It can be observed from the above analysis that there was no significant difference in the number of DEGs in the shoots and roots of the nitrogen-efficient genotype and the nitrogen-sensitive genotype.

#### 2.4.3. GO Enrichment Analysis of DEGs

To understand the function of differential genes, GO enrichment analysis was performed for the leaves and roots of W20 and W26. The numbers of DEGs in the leaves of W26 and W20 under different treatments were 3630 and 4358, respectively, and those in the roots were 4728 and 4518, respectively. These DEGs could be divided into three categories by GO enrichment analysis, namely, biological process, molecular function, and cell component (cellular components). The first 50 terms, significantly enriched according to the results of *p*_adj_ ≤ 0.05 GO enrichment analysis, were analyzed with column charts drawn.

According to the GO enrichment analysis of leaves and the classification of the “biological process”, the six most common functional groups (Appendix A) enriched by W26 and W20 were “cellular homeostasis”, “cell redox homeostasis”, “nucleoside metabolic process”, “glycosyl compound metabolic process”, “metal ion transport”, and “alpha-amino acid metabolic process”. The specific functional groups of W26 were “cellular amino acid metabolic process”, “ncRNA metabolic process”, “tRNA metabolic process”, “response to biotic stimulus”, and “protein folding”. The specific functional groups of W20 were the 11 functional groups, e.g., “multi-organism process”, “defense response”, “cell recognition”, “pollination”, and “pollen-pistil interaction”. As for the “cell components”, the functional groups enriched by the two varieties were the same, namely, “photosystem II”, “photosystem II oxygen evolving complex”, and “oxidoreductase complex”. There were 20 common functional groups in the classification of “molecular function”, such as “calcium ion binding”, “gated channel activity”, “ion gated channel activity”, “carbohydrate binding”, “pattern binding”, and “polysaccharide binding”. The specific functional groups of W26 included the 10 functional groups, such as “ligase activity”, “catalytic activity, acting on a tRNA”, “nucleoside binding”, “purine nucleoside binding”, “GTP binding”, and “ribonucleoside binding”. There specific functional groups of W20 included the eight functional groups, such as “substrate-specific channel activity”, “transferase activity, transferring hexosyl groups”, “sequence-specific DNA binding”, “oxidoreductase activity, acting on single donors with incorporation of molecular oxygen”, “ADP binding”, and “signaling receptor activity”.

According to the GO enrichment analysis of roots DEGs, W26 and W20 had seven common functional groups (Appendix A) in the classification of “biological processes”, namely, “response to oxidative stress”, “response to biotic stimulus”, “amine biosynthetic process”, “cellular biogenic amine biosynthetic process”, “multi-organism process”, “cellular biogenic amine metabolic process”, and “cellular amine metabolic process”. W26 had eight specific functional groups, e.g., “tricarboxylic acid metabolic process”, “nicotianamine metabolic process”, “nicotianamine biosynthetic process”, “tricarboxylic acid biosynthetic process”, “oxoacid metabolic process”, and “organic acid metabolic process”. W20 had the specific functional groups, e.g., “cell recognition”, “pollination”, “pollen-pistil interaction”, “recognition of pollen”, and “reproduction”. As for the “cell components”, the functional groups enriched by the two varieties were the same, namely, “extracellular region”, “cell wall”, “external encapsulating structure”, “apoplast”, and “cell periphery”. In the “molecular function” category, the two varieties shared the main functional groups of “peroxidase activity” and “oxidoreductase activity”, including 15 functional groups, such as “oxidoreductase activity, acting on peroxide as acceptor”, “antioxidant activity”, “ADP binding”, “transferase activity, transferring hexosyl groups”, and “hydrolase activity, hydrolyzing O-glycosyl compounds”. The specific functional groups of W26 were “nicotianamine synthase activity”, “sulfotransferase activity”, “transferase activity, transferring sulfur-containing groups”, “oxidoreductase activity, acting on single donors with incorporation of molecular oxygen, incorporation of two atoms of oxygen”, “ligand-gated ion channel activity”, and “ligand-gated channel activity”. The specific functional groups of W20 were the six functional groups, namely, “carbohydrate binding”, “transferase activity, transferring acyl groups”, “coenzyme binding”, “glucosyltransferase activity”, “transferase activity, transferring acyl groups other than amino-acyl groups”, and “vitamin binding”.

According to the GO enrichment analysis, W26 and W20 had both common and specific functional groups in the classification of biological processes and molecular functions. However, in the classification of cell components, the greatly enriched functional groups of the roots of the two genotypes were completely the same.

#### 2.4.4. KEGG Enrichment Analysis of DEGs

The Kyoto Encyclopedia of Genes and Genomes (KEGG) is a comprehensive database that integrates genomic, chemical, and system function information. In the study, KEGG analyzed the pathway enrichment in the roots and leaves of the two genotypes with different NUEs, with *p*_adj_ as the threshold of significant enrichment. According to the analysis of leaves, both W26 and W20 have 114 pathways involved. According to the KEGG analysis of roots, 111 and 112 pathways were identified in W26 and W20, respectively. Among the pathways of significant enrichment in leaves, there were 10 and 14 pathways in W26 and W20, and 5 and 8 in roots (*p*_adj_ ≤ 0.05). All significant enrichment pathways are related to carbon metabolism, nitrogen metabolism, fatty acid synthesis, and flavonoid biosynthesis.

According to the results of KEGG significant enrichment in leaves, there were 10 pathways (Figure 5a,b) shared by the two genotypes, namely, “Ribosome”, “Photosynthesis-antenna proteins”, “Glyoxylate and dicarboxylate metabolism”, “Photosynthesis”, “Porphyrin and chlorophyll metabolism”, “Carbon metabolism”, “Biosynthesis of cofactors”, “Carbon fixation in photosynthetic organisms”, “Aminoacyl-tRNA biosynthesis”, “Glycine, Serine, and threonine metabolism”. The specific enrichment pathways of W20 were “Glutathione (GSH) metabolism”, “amino acid biosynthesis”, “Pentose phosphate pathway”, and “Carotenoid biosynthesis”.

As for the common pathways of roots KEGG enrichment (*p*_adj_ ≤ 0.05) were four pathways shared by the two genotypes (Figure 5c,d), namely, “Photosynthesis-antenna proteins”, “Phenylpropanoid biosynthesis”, “Nitrogen metabolism”, and “Photosynthesis”. The specific enrichment result of W26 was “Plant-pathogen interaction”, and the specific enrichment results of W20 were “Flavonoid biosynthesis”, “Glyoxylate and dicarboxylate metabolism”, “Cysteine and methionine metabolism”, and “Carbon metabolism”.

#### 2.4.5. Validation of DEGs by Quantitative Real-Time PCR (qRT-PCR)

To verify the results of RNA-seq, qRT-PCR was used to analyze the expression of five genes in the leaves and roots of the two genotypes, respectively. The qRT-PCR analysis results were basically consistent with the RNA-seq data (Appendix A). These results confirm the reliability of the RNA-seq results and reflect the actual transcriptome changes in this study.

### 2.5. Metabonomics and Correlation Analysis

#### 2.5.1. Selection of DAMs under Different Nitrogen Treatments

After 3 days of low nitrogen, a total of 421 DAMs were identified in W26 leaves (including 172 up-regulated DAMs and 249 down-regulated DAMs), and 463 were identified in W20 leaves (including 287 up-regulated and 176 down-regulated) (Figure 6a–d). On the 18th day, 367 DAMs were identified in W26 (including 202 up-regulated and 165 down-regulated), while 240 DAMs were identified in W20 (including 85 up-regulated and 155 down-regulated). On the 21st day, 185 DAMs were identified in W26 (including 61 up-regulated and 124 down-regulated), while 83 DAMs were identified in W20 (including 20 up-regulated and 63 down-regulated).

As for the DAMs in the roots of the two genotypes, a total of 355 DAMs were identified in W26 after 3 days (including 176 up-regulated and 179 down-regulated), while 243 DAMs were identified in W20 (including 76 up-regulated and 167 down-regulated) (Figure 6e–h). After 18 days, there were 436 DAMs identified in W26, (including 310 up-regulated and 126 down-regulated), and 387 DAMs in W20 (including 202 up-regulated and 185 down-regulated). On the 21st day, 298 DAMs were identified in W26 (including 79 up-regulated and 219 down-regulated), while 227 DAMs were identified in W20 (including 85 up-regulated and 142 down-regulated).

As for the 3 days, with exception of the 3rd day when the DAMs in leaves W26 were less than those of W20, the total numbers of DAMs in W26 were much higher than those in W20 at the other time points, as well as the DAMs in the roots of W26 on the 3rd day. This indicated that the nitrogen-efficient material W26 had a much stronger response to low nitrogen stress. There were also 23 co-expressed DAMs identified in the experiment, including 14 in leaves and 9 in roots.

In addition, in the identification of differential metabolites, one plant hormone was identified as 3-indole butyric acid (IBA) in leaves, and two hormones were identified as 3-indolepropionic acid and brassinolide (BR) in roots. Two candidate hormones with low stress tolerance were selected from the roots (Figure 7).

#### 2.5.2. KEGG Joint Analysis of DAMs and DEGs

By mapping the DEGs and DAMs to the KEGG pathway database, their common pathway information was obtained, and the main biochemical pathways and signal transduction pathways involved in DAMs and DEGs were determined [41]. There were 26 KEGG co-enrichment pathways in W26 (Figure 8a), and GSH metabolism was a pathway that was significantly enriched between DAMs and DEGs. Thirty co-enrichment pathways were identified in W20 by KEGG joint analysis (Figure 8b), and GSH metabolism was also significantly enriched. Through analysis of the metabolic pathway, major metabolites were also found. DAMs, GSH, amino acids, and amides were the main identified DAMs in leaves.

According to the KEGG joint analysis of roots between DEGs and DAMs, 30 co-enrichment pathways were identified in W26 (Figure 8c), among which phenylpropane biosynthesis was the significantly different metabolic pathway between DAMs and DEGs. Twenty-eight co-enrichment pathways were identified in W20 (Figure 8d), and GSH metabolism was also significantly enriched between DAM and DEGs. It indicated that GSH metabolism was most closely related to low nitrogen in barley. In addition, the enrichment results showed that GSH, amino acid, and phenylpropane were the main DAMs found in roots.

### 2.6. Differential Genes and Metabolites of Nitrogen Metabolism and GSH Metabolism

Under low nitrogen stress, barley absorbed nitrate and ammonium, and completed the basic metabolic of nitrogen by virtue of a series of transporters and related enzymes. Appendix A exhibit a list of related differential genes in nitrogen metabolism identified in this study. Figure 9 presents the nitrogen metabolism pathway and the associated differential gene heat chart, in which those marked with yellow dots are the similar upward and downward trends of W26 and W20 at the three different time points. In leaves, the gene HORVU.MOREX.r3.6HG0541410 controlled the nitrate reductase. In roots, nitrate transporter genes were HORVU.MOREX.r3.6HG0543390 and HORVU.MOREX.r3.6HG0543380, ammonium transporter was HORVU.MOREX.r3.5HG0530810, nitrate reductase gene was HORVU.MOREX.r3.6HG0541410, and glutamine synthetase gene was HORVU.MOREX.r3.6HG0613270. The aforementioned genes with similar expression trends in the two genotypes can be considered the core genes in the nitrogen metabolism pathway in this study, having no concern with the genotypes. On the way, the other DEGs were the genes with different expression trends in the two genotypes.

As shown in Figure 9, glutathione metabolism and nitrogen metabolism can be connected. For GSH metabolism of leaves, HORVU.MOREX.r3.7HG0713570, HORVU.MOREX.r3.1HG0051860, and HORVU.MOREX.r3.7HG0666960 genes were down-regulated and up-regulated in two genotypes, respectively, and they all belonged to GSH S-transferase. A gene with the same expression trend was identified in roots. HORVU.MOREX.r3.2HG0188850 belonged to GSH peroxidase. Simultaneously, five different metabolites were identified, oxidized GSH (GSSG), cysteinylglycine (Cys-Gly), L-γ-glutamyl-L-amino acid (L-γ-glutamyl-L-aminoacid), and 5-oxypropane (5-oxoproline). The upward and downward trends of 5-oxypropane (5-oxoproline) and (5-murine L-glutamyl)-L-amino acids in leaves and L-γ-L-glutamyl-L-amino acids (Glutamyl-L-aminoacid) in roots were the same.

## 3. Discussion

Nitrogen is an essential nutrient element that plays an important role in plant growth and development. In this study, the biomass and nitrogen content of the two materials decreased after low nitrogen. According to previous studies, low nitrogen stress can inhibit plant growth and reduce shoot dry weight and total biomass [42,43]. Biomass or dry weight can usually be used as an index of plant tolerance under nutritional stress [44,45]. When applying low nitrogen stress, plants may also absorb more nitrogen by increasing the root–shoot ratio, so as to cope with the low nitrogen stress [43]. As for the barley genotypes with a high NUE, the total biomass and nitrogen accumulation decreased after low nitrogen stress, but the NUE increased [46]. In this study, by measuring the dry weight and nitrogen content of the genotype, it was proved that W26 was more tolerant to low nitrogen and had a higher NUE than W20.

The related genes of the plant nitrogen metabolism pathway have a close relationship with plant NUE. Nitrate is the main form of nitrogen absorbed by plant roots, and the nitrate transporter (*NRT*) is mainly responsible for nitrate transport. For example, *NRT1.1* in *Arabidopsis thaliana* is a parental nitrate transporter that can absorb nitrate over a wide range of concentrations [47]. Moreover, the *NRT1*/*PTR* of proton-coupled transporters is responsible for nitrogen assimilation in eukaryotes and bacteria, and members of this family have evolved to transport nitrates and other secondary salts in most plant species [48]. In *Arabidopsis thaliana AtNRT1.1* knockout mutants with high nitrogen levels, the expression levels of high-affinity nitrogen transporter genes, such as *AtNRT2.1*, *AtNRT2.4*, and *AtAMT1*, showed a decrease [49,50]. The ammonium transporter (*AMT*) represents the main entry pathway of NH_4_^+^ absorbed by roots. As excessive ammonium absorbed by plants may lead to poisoning, ammonium absorbed by the root plasma membrane must be strictly regulated [51]. In addition to nitrate transport proteins and ammonium transport proteins, relevant enzymes also play an important role in plant nitrogen metabolism pathways. For example, nitrate reductase (NR) activity can affect plant NUE. Scholars have found that mutated indica rice and japonica rice have different nitrogen use efficiencies, and this is due to the different nitrate reductase activities. The variation of *OsNR2* alleles encoding nitrate reductase results in OsNR2 proteins with different alleles encoding structures of mutated indica rice and japonica rice, while indica rice *OsRN2* shows higher NR activity [52]. Low nitrogen stress can cause a great decrease in the transcription levels and activities of NR, NIR, GS, and GOGAT. For example, after applying low nitrogen stress, and treating wheat with potassium nitrate and ammonium nitrate, the expression and activity of NR, NIR, GS, and GOGAT are restored [53]. According to studies, glutamine synthetase 2 (GS2) and FD-GOGAT are two key enzymes involved in the GS/GOGAT cycle, which are necessary for plant nitrogen assimilation [53]. The GS-GOGAT pathway is the key metabolic pathway of glutamate and glutamine. By optimizing this pathway, the metabolic flux of glutamine can be caused, thereby increasing the production of glutamine and reducing the production of by-product glutamate [54]. Glutamine and glutamate are metabolized to aspartate and asparagine by aspartate aminotransferase and asparagine synthetase, respectively [55].

As the signaling molecules of low nitrogen stress, plant hormones have a complex regulatory network under low nitrogen stress. IBA plays a strong role in all aspects of root development, including the regulation of root tip meristem size, root hair elongation, lateral root development, and adventitious root formation [56]. Studies on maize indicate that BR treatment can increase the biomass and nitrogen yield index [57]. The application of BR can greatly increase the chlorophyll content, photosynthetic rate, and light energy use efficiency of seedlings, and promote the activities of key enzymes in nitrogen metabolism [58].

In this study, GSH metabolism was the pathway of significant enrichment of the two genotypes in the KEGG joint analysis. In a *Saccharomyces cerevisiae* study, the content of GSH (GSH) increased from 7 to 17 nmol (mg dry weight)^−1^ during the first 2 h [59]. The total nitrogen content of soybean treated with Ag-NP (which can inhibit the formation of nodules) and GSH was more than 5 times higher than that of soybean treated with Ag-NP alone [60]. The above studies show that low nitrogen stress can induce an increase in GSH content, and GSH also increases the accumulation of nitrogen. GSH was the most important antioxidant that regulated plant abiotic stress response [61,62]. It could also stabilize the intracellular redox dynamic balance, stimulate stress-related signals, detoxify foreign substances, and promote stress survival [63]. Under the low nitrogen treatment of *Labisia pumila* Blume, it was found that the antioxidant activities (DPPH and FRAP) were significantly positively correlated with total flavonoids, GSH, GSSG, anthocyanins, and ascorbic acid, indicating that the higher content of these compounds under low nitrogen conditions might be one of the reasons for the enhanced antioxidant activity [64]. GSH was oxidized to GSH-disulfide in plant cells, and performed normal physiological functions under stress. GSH was also a reservoir of reduced sulfur, and played a vital role in nucleic acid and protein synthesis that regulated enzyme activity [64]. GSH repairs -SH groups through a GSH-disulfide exchange reaction to protect the -SH groups of some enzymes and structural proteins from oxidation [65]. Previous studies have shown that some *APX*, *GPX*, and *GST* genes are induced under oxidative stress [66]. In eukaryotic cells and almost all Gram-negative bacteria, GSH synthase (GSH2) and γ-glutamylcysteine synthase (GSH1) catalyze GSH synthesis, and GSH1 is inhibited by GSH feedback [67]. It was found in maize that GGT activity and protein were unevenly distributed in tissues, and were more distributed in the epidermis and stomata, parenchyma of conducting elements, and root meristem [68]. The above studies show that GSH can indeed play a role in the response of plants to low nitrogen stress, and this role may be related to GSH’s antioxidant protection of nucleic acid and protein activities.

## 4. Materials and Methods

### 4.1. Plant Growing and Sampling

The full seeds of barley varieties W26 and W20 were soaked in 5% sodium hypochlorite solution for 10 min, rinsed three times, and germinated in a germination box with three layers of filter paper, ensuring an appropriate amount of sterilized water was sprayed every day. Seven days later, seedlings with the same growth were selected and transferred to hydroponic boxes. The hydroponic box has a volume of 10 L and was used with Hoagland’s nutrient solution. Normal concentration was set to 2 mM [24] (Table 2; Nitrogen concentration of CK and RN), and the low nitrogen concentration was set to 0.4 mM (LN). Since the seedlings were transplanted to the hydroponic box with normal nitrogen treatment and low nitrogen treatment, the nutrient solution needed to be renewed every 6 days. When the seedlings grew to the 18th day, half of the seedlings treated with low nitrogen began to be treated with normal nitrogen (RN) (Figure 1a), and the other half was subjected to continuous low nitrogen, with the other managements the same. During the growth of seedlings, the number of plants in each hydroponic box is 12. Seedlings were sampled on the 3rd, 18th, and 21st days. Then, the samples were rapidly frozen in liquid nitrogen and stored at −80 °C for later RNA-seq and metabolite analysis. Moreover, some seedings were separated into the shoots and the roots, which were then placed at 105 °C for 40 min and dried at 70 °C to obtain a stable weight for dry weight determination, and 3 biological repeats were necessary. Nitrogen content was determined with BUCHIKjelMasterK-375.

### 4.2. Extraction of Total RNA and the Construction and Sequencing of RNA-Seq Library

All samples (in total, 72, 2 genotypes (W26 and W20) × 2 parts (leaves and roots) × 2 treatments (LN and CK/RN) × 3 time points (3rd day, 18th day, and 21st day) × 3 biological replications) were prepared for further RNA-seq analysis. After the total RNA was extracted with the Radix Scutellariae polysaccharide polyphenol kit DP441, the concentration, purity, and integrity of RNA were determined by adopting the Nano Photometer^®^ spectrophotometer, the Qubit^®^ 2.0 fluorometer, and Agilent 5400, respectively. Finally, the RNA samples, whose RIN values were greater than 7, were selected to build a database for sequencing. The mRNA with a poly-A tail was enriched by total RNA with Oligo (dT) magnetic beads, and then the mRNA was randomly interrupted by divalent cations in fragmentation buffer and used as a template. Random oligonucleotides were used as primers to synthesize the first chain of cDNA in the M-MuLV reverse transcriptase system, and then RNaseH was used to degrade the RNA chain. The second chain of cDNA was synthesized with dNTPs in the DNA polymerase I system. After giving terminal repair, A tail addition, and sequencing junction connection to the purified double-stranded cDNAs, the cDNAs of about 370–420 bp were selected with AMPure XP beads, the PCR amplification was performed, and the PCR product was purified with AMPure XP beads. Finally, the library was obtained.

After completing the construction of the library, the Qubit 2.0 fluorometer was used to quantify the library initially, and then the library was diluted to 1.5 ng/uL. Next, the insert size of the library was detected by using the Agilent 2100 bioanalyzer. After the insert size met the expectations, the effective concentration of the library needed to be measured accurately (the effective concentration of the library was higher than 2 nM) with qRT-PCR to ensure the quality of the library.

After passing the library inspection, different libraries were sequenced by using Illumina NovaSeq 6000 (Illumina, San Diego, CA, USA) after pooling, according to the effective concentration and the target amount of off-machine data, with 150 bp paired-end reads produced. The basic principle of sequencing was sequencing while synthesizing (sequencing by synthesis). Four kinds of fluorescently labeled dNTP, DNA, polymerase, and splice primers were added to the sequenced flow cell for amplification. When extending each sequenced cluster with the complementary chain, each fluorescently labeled dNTP can release the corresponding fluorescence. The sequencer could capture the fluorescence signal, and the optical signal could be converted into the sequencing peak by computer software, thus obtaining the sequence information of the fragment to be tested.

### 4.3. Sequencing Data Quality Control

The image data measured by the high-throughput sequencer were transformed into sequence data (reads) by CASAVA base recognition. The file was in fastq format, and it mainly contained the sequence information of sequencing fragments and the corresponding sequencing quality information. The original data obtained by sequencing included a small amount of reads with sequencing connectors or low sequencing quality. To ensure the quality and reliability of data analysis, it was necessary to filter the original data, mainly including the removal of reads with connectors (adapter), the removal of reads containing N (N means that the base information cannot be determined), and the removal of low quality reads (reads whose base number of Q_phred_ ≤ 20 accounted for more than 50% of the total read length). Simultaneously, the contents of Q20, Q30, and GC in clean data were calculated, and all subsequent analyses were based on the high-quality clean data analysis.

The reference genome (*Hordeum_vulgare*_MorexV3) and gene model annotation files can be downloaded directly from the website “https://ftp.ensemblgenomes.ebi.ac.uk/pub/plants/release-54/fasta/hordeum_vulgare/dna/ (accessed on 19 August 2022)”. HISAT2v2.0.5 was used to build an index of the reference genome, and HISAT2v2.0.5 was adopted to compare the paired terminal clean reads with the reference genome. HISAT2 was selected as the alignment tool, because HISAT2 could generate spliced databases based on the gene model annotation files, and it had better alignment results than other non-splicing comparison tools.

### 4.4. Identification and Analysis of Metabolites

Six biological repeats are required in the identification of metabolites, so a total of 144 samples are used for analyses. The Vanquish UHPLC liquid chromatograph and the QExactive liquid HF-X liquid phase mass spectrometer were used in scanning the prepared metabolite extracts and QC samples (QC samples were equal volume mixed samples of the experimental samples, used to balance the GC-MS system and monitor the status of the instrument, and to evaluate the stability of the system during the whole experiment). The data measured with the liquid phase mass spectrometer were preprocessed by CD3.1 data processing software, to make the identification more accurate. Then, the peak was extracted according to the ppm, signal–noise ratio, and adduct ion set, as well as other information, and the peak area was quantified simultaneously. Next, the metabolites were identified by comparing the high-resolution secondary spectrogram databases mzCloud and mzVault and the first-level database of MassList (searching the database), with the specific principles as follows. The molecular weight of the metabolite was determined according to the mass–charge ratio of the parent ion in the first-order mass spectrometry, the molecular formula was predicted with information such as mass number deviation (ppm) and adduct ions, and then the database was matched. Moreover, the database with the secondary spectrum matched the information of fragment ion and collision energy of each metabolite in the database, according to the actual secondary spectrum, thereby realizing the secondary identification of metabolites. The metabolites, with a coefficient of variation less than 30% in QC samples [69], were retained as the final identification results for subsequent analysis.

### 4.5. Selection of DEGs and DAMs

In the analysis of leaves and roots, LN/CK was taken as the comparison pair of the 3rd and 18th days, and LN/RN was taken as the comparison pair of the 21st day.

DESeq2 software was used in standardizing the original read count and detecting the differentially expressed genes, and then |lg(FoldChange)| ≥ 1 & *p*_adj_ ≤ 0.05 was adopted as the standard for selecting DEGs.

The screening of DAMs mainly referenced the three parameters, namely, VIP, FC, and *p* value. VIP refers to the variable importance in the projection of the first principal component of the PLS-DA model [70], with the value indicating the contribution of the metabolite to the grouping. Fold change refers to the multiple of differences, which was the ratio of the mean value of the repeated quantitative values of all organisms in the comparison group. *p* value was calculated by T-test [71], indicating the significant level of difference. The threshold was set as VIP ≥ 1.0, lg(FoldChange)| ≥ 1 & *p* value ≤ 0.05 [70,72,73].

### 4.6. qRT-PCR

To verify the reliability of the results of leaves and roots transcriptome sequencing, qRT-PCR was carried out on 5 genes related to low nitrogen in roots and leaves, respectively. The first strand of cDNA was synthesized by the Goldenstar RT6 cDNA Synthesis Mix (Tsingke, Wuhan, China) kit, and qRT-PCR was carried out by using instrument QuantStudioTM 1 Plus (ABI, Carlsbad, CA, USA) and 2 × T5 Fast qPCR Mix (SYBR Green I) (Tsingke, Wuhan, China) kits. The relative expression of each gene was calculated by the 2^–ΔΔC^ method [74], and the internal reference was *HvActin* (NCBI key number: AY145451) [75].

## 5. Conclusions

As indicated by the analyses of RNA-seq and metabolites of W26 and W20 under low nitrogen, there were great differences in LN and RN among barley genotypes with different NUEs. Under low nitrogen, the differential genes and DAMs of the two genotypes were obviously enriched in GSH metabolism, which could be related to the regulation of GSH. The transporters of the *NRT* and *AMT* genes; the *NR*, *NIR*, *GS*, and *GDH* genes; and the *GOGAT* genes were also selected in the main pathways of nitrogen metabolism, including the genes of tolerance to low nitrogen stress. Among them, some of the genes had no concern with varieties, showed the same upward and downward trends in the two genotypes, and could also be called core genes of tolerance to low nitrogen. This study provides a theoretical basis for further understanding the complex metabolic process of barley under low nitrogen stress. The functional verification of candidate genes for nitrogen-efficient utilization will continue to be carried out in the future, which will improve NUE in crops.

## Figures and Tables

**Figure 1 ijms-24-04706-f001:**
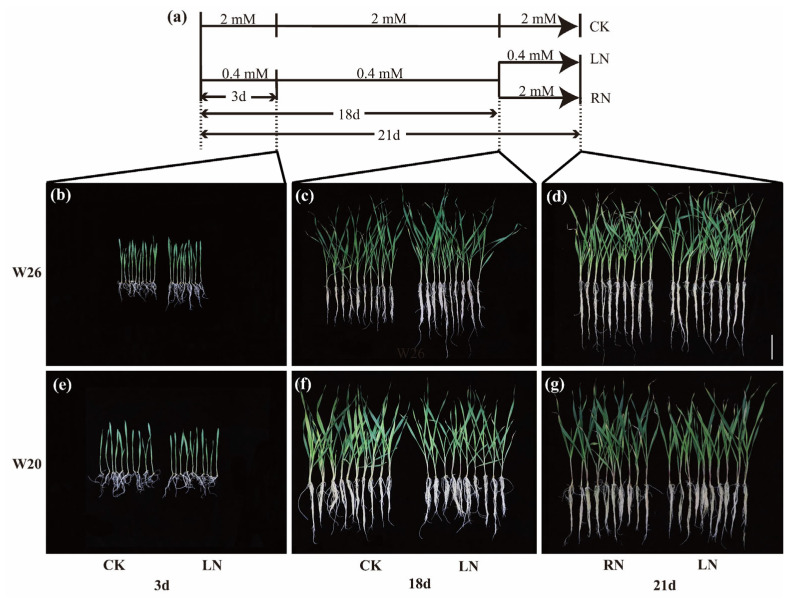
The experimental treatment and the growth of barley at different sampling times. CK represents normal nitrogen, LN represents low nitrogen, and RN represents nitrogen resupply after low nitrogen stress. (**a**) Schematic diagram of different treatments in the experiment. (**b**–**d**) Growth charts of W26 on the 3rd, 18th, and 21st days. (**e**–**g**) Growth charts of W26 on the 3rd, 18th, and 21st days. The segment at the lower right corner is the scale of the six drawings, representing the length of 10 cm.

**Figure 2 ijms-24-04706-f002:**
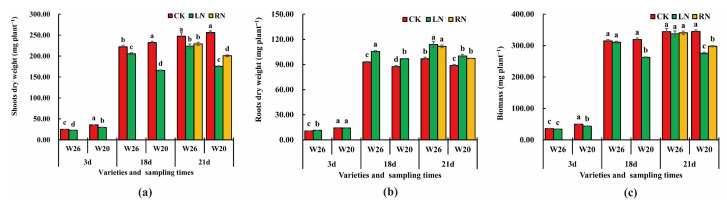
Barley dry weight and biomass at the three time points under different nitrogen treatments. CK represents normal nitrogen, LN represents low nitrogen, and RN represents nitrogen resupply after low nitrogen stress. (**a**) Shoot dry weight; (**b**) root dry weight; (**c**) biomass. The small letters represent the significant difference at the same time (*p* < 0.05).

**Figure 3 ijms-24-04706-f003:**
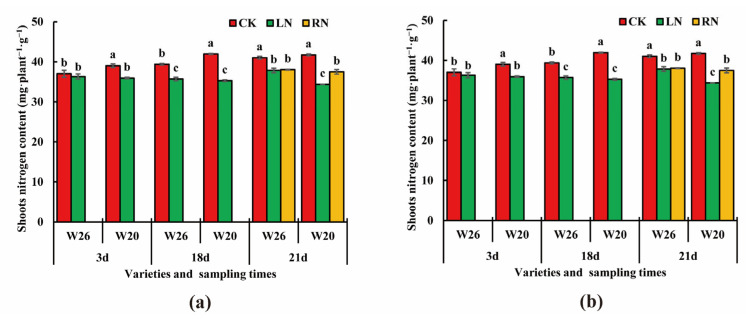
Nitrogen content at the three time points under different nitrogen treatments. CK represents normal nitrogen, LN represents low nitrogen, and RN represents nitrogen resupply after low nitrogen stress. (**a**) Nitrogen content of shoots; (**b**) nitrogen content of roots. The small letters represent the significant difference at the same time (*p* < 0.05).

**Figure 4 ijms-24-04706-f004:**
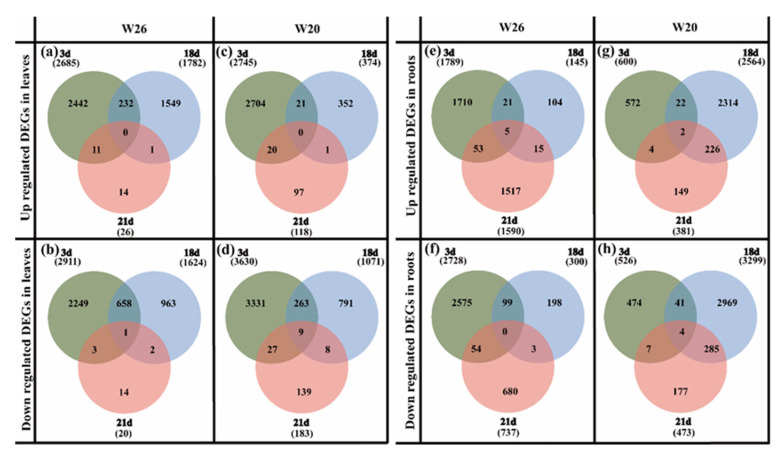
Venn diagram of DEGs W26 and W20 at different time points. (**a**,**b**) show the number of up-regulated differentially expressed genes (DEGs) and down-regulated DEGs in W26 leaves, respectively. (**c**,**d**) show the number of up-regulated DEGs and down-regulated DEGs in W20 leaves, respectively. (**e**,**f**) show the number of up-regulated DEGs and down-regulated DEGs in W26 roots, respectively. (**g**,**h**) show the number of up-regulated DEGs and down-regulated DEGs in W20 roots, respectively.

**Figure 5 ijms-24-04706-f005:**
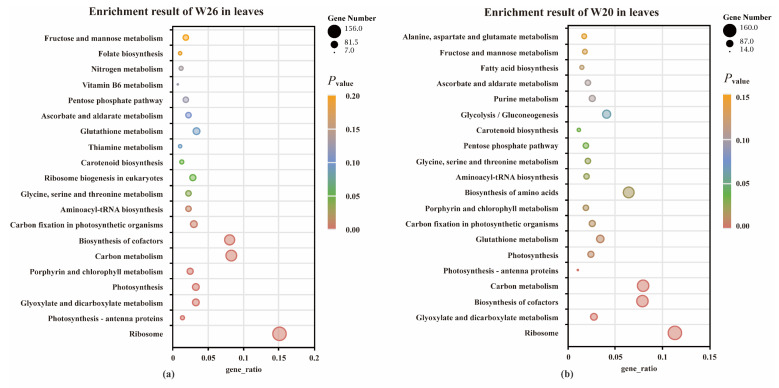
The statistics of KEGG enrichment in leaves and roots of barley. (**a**,**b**) show the KEGG enrichment results of the differentially expressed genes (DEGs) in W26 and W20 leaves, respectively. (**c**,**d**) show the KEGG enrichment results of the DEGs in W26 and W20 roots, respectively. The abscissa in the figure is the ratio of the number of DEGs annotated on the KEGG pathway to the total number of DEGs, and the ordinate is the KEGG pathway name. The depth of color represents the degree of enrichment.

**Figure 6 ijms-24-04706-f006:**
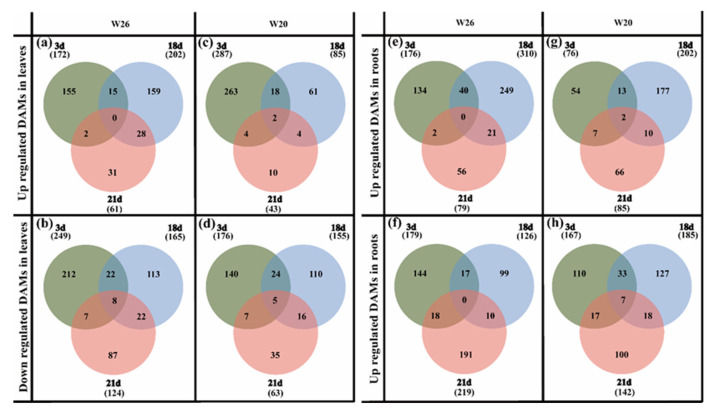
Venn diagram of DAMs W26 and W20 at different time points. (**a**,**b**) show the number of up-regulated differential metabolites (DAMs) and down-regulated DAMs in W26 leaves, respectively. (**c**,**d**) show the number of up-regulated DAMs and down-regulated DEGs in W20 leaves, respectively. (**e**,**f**) show the number of up-regulated DAMs and down-regulated DAMs in W26 roots, respectively. (**g**,**h**) show the number of up-regulated DEGs and down-regulated DAMs in W20 roots, respectively.

**Figure 7 ijms-24-04706-f007:**
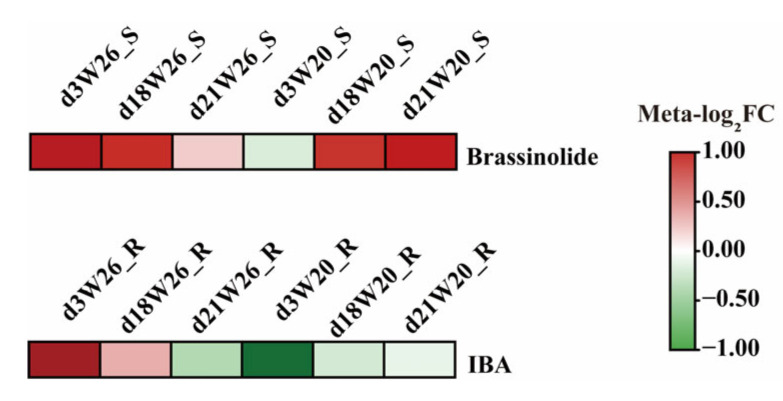
Heat map of relative expression of plant hormones. “d3W26_S” represents the leaves sampled by W26 on the 3rd day. “d18W26_S” represents the leaves sampled by W26 on the 18th day. “d21W26_S” represents the leaves sampled by W26 on the 21st day. “d3W26_R” represents the roots sampled by W26 on the 3rd day. “d18W26_S” represents the roots sampled by W26 on the 18th day. “d21W26_S” represents the roots sampled by W26 on the 21st day. “Meta” represents metabolite. “FC” represent “FoldChange”.

**Figure 8 ijms-24-04706-f008:**
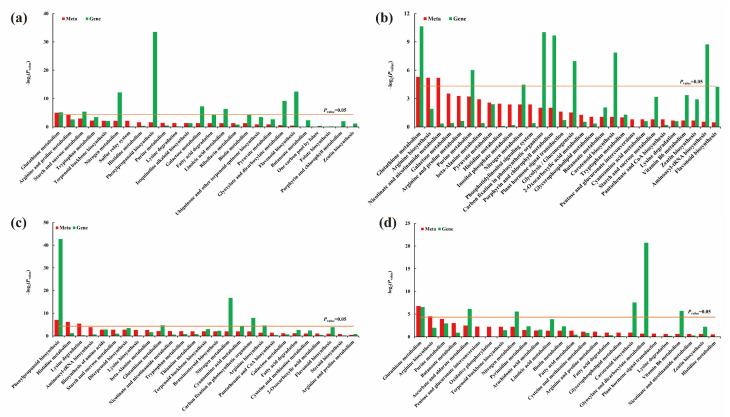
(**a**) Histograms of joint KEGG enrichment *p* value between DAMs and DEGS in leaves of W26; (**b**) histograms of joint KEGG enrichment *p* value between DAMs and DEGs in leaves of W20; (**c**) histograms of joint KEGG enrichment *p* value between DAMs and DEG_S_ in roots of W26; (**d**) histograms of joint KEGG enrichment *p* value between DAMs and DEG_S_ in roots of W20. “Meta” represents KEGG pathway enriched by DAMs, and “Gene” represents KEGG pathway enriched by DEGs. There is a significant difference above the yellow line (*p* value < 0.05).

**Figure 9 ijms-24-04706-f009:**
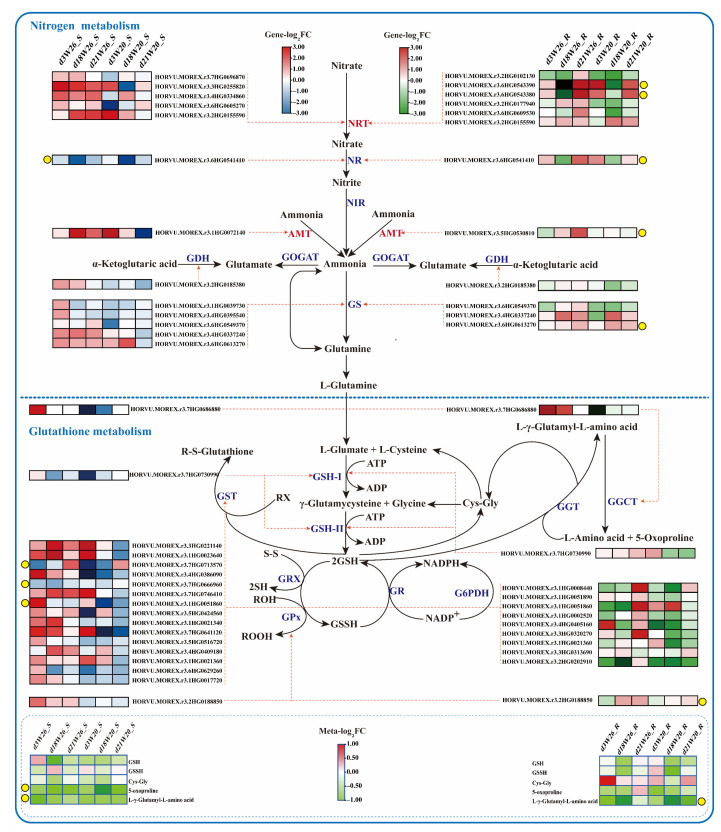
Nitrogen metabolism and GSH metabolism pathways. NR, nitrate reductase; NIR, nitrite reductase; GOGAT, glutamate synthetase; GDH, glutamate dehydrogenase; GS, glutamine synthetase; NRT, nitrate transporter; AMT, ammonium transporter; GSH-1, γ-Glutamyl cysteine synthetase; GSHII, glutathione synthetase; ATP, adenosine triphosphate; S-S, disulfide bond; SH, sulfhydryl; ADP, adenosine diphosphate; RX, organic halide; GST, glutathione S-transferase; GRX, glutaredoxin; GPx, glutathione peroxidase; ROOH, hydroperoxide; ROH, hydroxide; G6PDH, 6-phosphate glucose dehydrogenase; GR, glutathione reductase; GSH, glutathione (reduced glutathione); GSSH, oxidized glutathione; Cys-Gly, cysteine glycine; GGT, glutamyltransferase; R-S-glutathione, glutathione thioate; GGCT, γ-Glutamic acyltransferase. FC (FoldChange) is the multiple of relative expression. “Meta” stands for metabolite. The small square in the figure represents the relative expression of genes or metabolites in W26 and W20. The red word represents the transporter. The yellow dot in the figure represents that the relative expression trend of the gene or metabolite at the corresponding sampling time in the two genotypes is similar. “d3W26_S” represents the leaves sampled by W26 on the 3rd day. “d18W26_S” represents the leaves sampled by W26 on the 18th day. “d21W26_S” represents the leaves sampled by W26 on the 21st day. “d3W26_R” represents the roots sampled by W26 on the 3rd day. “d18W26_S” represents the roots sampled by W26 on the 18th day. “d21W26_S” represents the roots sampled by W26 on the 21st day. “Meta” represents metabolite. “FC” represent “FoldChange”.

**Table 1 ijms-24-04706-t001:** Nitrogen use efficiency (NUE) after 21 days of plant growing.

Genotype	Treatment	NUE (%)
W26	CK	19.50 c
	LN	87.52 a
W20	CK	19.58 c
	LN	61.74 b

Note: CK is the normal nitrogen, LN is the low nitrogen treatment. Here, NUE was measured by nitrogen absorption efficiency. The small letters represent the significant difference (*p* < 0.05).

**Table 2 ijms-24-04706-t002:** Normal nitrogen nutrient solution formula.

Macronutrients		Micronutrients	
Name	Concentration (mM)	Name	Concentration (μM)
Ca(NO_3_)_2_·4H_2_O	0.53	KI	5
KNO_3_	0.67	H_3_BO_3_	100
NH_4_NO_3_	0.13	MnSO_4_	150
KH_2_PO_4_	0.13	ZnSO_4_·7H_2_O	30
MgSO_4_·7H_2_O	0.27	Na_2_MoO_4_·2H_2_O	1
		CuSO_4_·5H_2_O	0.1
		CoCl_2_·6H_2_O	0.1
		Na_2_EDTA·2H_2_O	20
		FeSO_4_·7H_2_O	20

Note: The concentration of Ca(NO_3_)_2_·4H_2_O, KNO_3_, and NH_4_NO_3_ in the nutrient solution of low nitrogen treatment decreased to 1/5 of the normal nitrogen, and insufficient K^+^ and Ca^2+^ were supplemented with CaCl_2_ and KCl.

## Data Availability

The transcriptome data of the 72 samples described in this study have been stored in the National Biotechnology Information Center (NCBI) database, with the biological project entry number PRJNA896249.

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
