# Peer review of "Transcriptome and Metabolome Reveal the Molecular Mechanism of Barley Genotypes Underlying the Response to Low Nitrogen and Resupply"

_ijms, 2023, doi:10.3390/ijms24054706_

Round 1

Reviewer 1 Report

In the current study, Transcriptome and metabolome analyses for two barleys with different nitrogen use efficiencies in low nitrogen stress and recovery explored some nitrogen efficient genes and metabolites. In addition, the authors also explored the pathway of transcription. The study topic is very interesting, novel, and helpful for the readerships of IJMS. The overall manuscript is well organized and well-written. However, the main concerns about this manuscript can be found below, and minor revision are suggested.

Major metabolites from roots and leaves must be specify in the abstract.

In abstract the authors should provide specific results. Current results are very generally explained.

Current and future perspective of the study should be provided in the abstract.

Line 75-76 add reference.

Line 89 to 97 the authors should provide proper literature review by discussing the mechanism and specific plants transcriptomic under N stress.

“Metabonomics” Line 98 check spelling.

Line 104 add reference.

Line 106-112 the author used word some studies but added only one reference.

Line 113 to 121 the justification is very weak and must be revised. Also provide specific objectives of the study.

Line 231 to 233 the sentence looks like suggestion to a plant. Revise the sentence and would be better to write specific results.

Gene names must be italicized in the whole MS.

Section 4.5 should be cited with relevant study. The following study could help the authors.

https://doi.org/10.1016/j.indcrop.2022.116090

There results of KEGG joint analysis of DAMs and DEGs but methodology is not clearly written. The authors are instructed to write the methodology and also cite relevant study. The following study could help the authors.

https://doi.org/10.3389/fgene.2021.635043

Add significance of the study in the conclusion. 

Reviewer 2 Report

·      What were the criteria to set low nitrogen stress for 3 days, and 18 days?

·      In line 21: it was metabonomic or metabolomic, please check.

·      In line 24, please check: were measures.

·      The keywords can be modified by removing the words from the title of the manuscript.

·      hoagland can be written as Hoagland’s. There are many more basic minor errors that need to be corrected, so please pay attention.

·      The glutathione regulation can be elaborated in the discussion part as it was one of the main concluding remarks of the study.

·      In conclusion, the application or research gap or future direction can be added to the conclusion part.

Reviewer 3 Report

Comments to the Authors

The manuscript “Transcriptome and metabolome analyses for two barleys with 2 different nitrogen use efficiencies in low nitrogen stress and recovery” has been revised. The authors supply a considerable amount of data, transcriptomic, and metabolomics to understand the whole plant performance of two barley cultivar differing in N use. Therefore, information extracted from these kind of studies results of high valuable interest.  Results are of great interest for their future applied perspectives, having in mind the huge amounts of N fertilizers needed today by agriculture to maintain crop production. However, the manuscript suffers from a lot of  weak points, that must be conscientiously revised and considered.

Title. It is recommended to focus it on a “conclusion” or “result”, when possible,  to make more specific and attractive, instead of only descriptive.

Introduction.

- Authors says “In the root, the nitrate is firstly restored to nitrite by the nitrate reductase (NR) in cytoplasm, and then further to ammonium by the nitrite reductase  (NIR) in chloroplast and the glutamine synthase (GS) in cytoplasm and cytoplasm”. How are these chloroplasts in the roots?. Please, the authors must check the partitioning of nitrate reduction process in root and leaves, since normally it is the leave the organ where the nitrate reduction process occurs in the majority of the plants (see Miflin, Nature 214, 1133–1134 (1967); Scheurwater et al., J Exp Bot. 2002. 53, 374, 1635-1642, doi.10.1093/jxb/erf008; and many others).

- Pay attention that, the alfa-ketoglurate is the C5-carbon skeleton that allows the ammonium assimilation, eventually by Glutamine synthetase and, probably, by Glutamate dehydrogenase (lines 70-71).

- Lines 113-114, It must be explained (and referenced) the backgrounds that lead to consider W26 and W20 cultivars/genotypes/varieties, etc (better terminology than materials) as different in nitrogen use efficiency. What is the nitrogen use efficiency for these cultivars?.

Material and Methods

- The specific nutrient solution (Hoagland) and composition must be described. The proportion of ammonium and nitrate is essential to interpret correctly the data obtained and to understand the N use by these cultivars.  As equally important, it is to specify the N source available during nitrogen recovery period.

- Define the nitrogen treatments in this section, with the respective abbreviations used thoroughly the work.

- It is said that the RIN number of RNA samples considerd to obtain RNA-Seq library is higher than 4. These vales seem too low, since normally values higher 7-8 are used. Please explain or revise it.

- Authors provide a lot data and instrumental about the procedures for metabolites and transcriptomic analysis. This is great, but it results too excessive and long. I would recommend to check, if possible, to reduce the extension.

Results

- Figure 3 must be presented firstly, previously biomass and N concentration in the plant.

- The N use efficiency could be calculated or estimated by the authors to provide different evidence of N use by these cultivars.

- Title of subsection 2.2 (lines 152) is repeated (the same as 2.1, lines 123)

- Line 153: specify that the results are describing the N content in the first sentence.

- The N concentrations units for macronutrients are normally expressed aspercentage, and not as units per thousands.

- Number of plants used for biomass and N concentration determination must be specified. These data are not clear enough: how many containers of plants were used, and the number of biological repetitions.

- Lines 182-185 must be rewritten to clarify the idea.

- The number of biological and technical replicates used for RNA-seq must be specified in Material and Methods. They are specified in “Section 2.3.2. Screening of DEG”, but are these details the same that for the previous section 2.3.1?. Similarly, information from 342-352 lines must be reconsidered to include in Material and Methods Section. The reorganization of this kind of information is needed, in order to understand in a logical manner the experimental procedure, and to follow more in depth the results obtained. And these information must be included in Materials and Methods more than in the Results section (lines 205-214). Otherwise, it is complicated to follow the main results and analysis properly.

- Line 243: What does it mean a “woody long distance” in a small gramineous seedling of barley?.  

- Lines 356-358, it must be specified the data refer to day 3rd.

-   Line 366: remove “of the two varieties”, since results correspond to W20.

- Line 370: value correct is 310

Line 384: What is 3-Indoleprofonic. Is it correct or a typing error?. IBA must be referred with the complete name the first time it is named.

- In Figure 9, it helps to mark at the top of the figure that the right side corresponds to the root and the leaf side to shoots.

Discussion

Lines 486-486: Asparagine is also formed from glutamine, and some times it is also probable asparagine increases more than glutamine. It must be taken into account when discussing this par.

- Lines 515: “methylethylation of glutathione” is correct? Or, does it refer to “methylation of proteins”?

 - See also below.

Overall comments:

The structure of the manuscript needs a revision, as commented, due to several items. The length of the manuscript results too excessive, with a detailed description “Material and Methods” and “Results”, but more poorly integrated the interesting results in the Discussion. As reader, one has the impression that authors focus and discuss much more the existing literature (on rice, on Corinebaterium bacteria, for instance) than focusing on their own results. For instance, the second paragraph in the Discussion refers to nearly all kinds or hormones, when basically only IBA (indole-3-butyric acid) and Brassinolide are quantitatively different in both cultivars. It should be more focused to the potential roles of brassinolide in shoots and that of IBA in roots.

There are a lot of information worthy to be discussed. The specific behavior for these cultivars regarding N use (by roots and shoot) should be a central part of the discussion. The role of glutathione is discussed, but accordingly to the data presented, the routes shown in Figure 9 should be discussed a bit more in depth, for the case of barley. For instance, the potential differences in the use of N sources (nitrate or ammonium) in the plant, according the induction of the routes or genes (NR, GRX, GR, etc) in both organs must be addressed. Enzymes for glutathione metabolism are differently expressed in leaves for both cultivars, being in general increased in W26, and depleted in W20 in leaves, and the same happens, overall, for GS.

The quantification of metabolites in tissues as nitrate, ammonium and amino acids, glutathione and protein, would help to understand better the differences in the use of N between both cultivars.
Can it be suggested that W26 be more efficient using ammonium or transporting N compounds to shoots?. Remember that this point can only be understood if we know the type of N source provided, and nitrogenous compounds in tissues.

Regarding the grammar, there are some confusing lines; sentences should be shorter, and some non essential information removed. A carefull revison is needed
